# Access to Services from Persons with Disabilities in Afghanistan: Is Community Based Rehabilitation Making a Difference?

**DOI:** 10.3390/ijerph19106341

**Published:** 2022-05-23

**Authors:** Jean-Francois Trani, Kyle A. Pitzer, Juanita Vasquez Escallon, Parul Bakhshi

**Affiliations:** 1Brown School, Washington University, St. Louis, MO 63130, USA; 2School of Medicine, Washington University, St. Louis, MO 63130, USA; kyleapitzer@wustl.edu; 3UNICEF—United Nations Children’s Fund, New York, NY 10017, USA; jvasquezescallon@unicef.org; 4Occupational Therapy Program, School of Medicine, Washington University, St. Louis, MO 63108, USA; bakhship@wustl.edu

**Keywords:** Afghanistan, community based rehabilitation, disability, difference in difference, propensity score matching

## Abstract

The United Nations Convention on the Rights of Persons with Disabilities (UNCRPD), ratified in 2006, states that the achievement of equal rights, empowerment, and social inclusion of people with disabilities requires comprehensive rehabilitation services involving educational, social, economic, and medical interventions, all dimensions of the World Health Organization Community based rehabilitation (CBR) matrix. CBR programs aim at achieving those goals. In the present study, we investigated whether a large scale CBR program is improving access to multiple services (namely physical therapy, assistive technology, education, employment, advocacy, and community awareness) and providing satisfactions (by measuring the reduction in unmet needs) of Afghans with disabilities. We enrolled in the study 1861 newly recruited CBR participants with disabilities from 169 villages between July 2012 and December 2013, and 1132 controls screened with disabilities randomly selected with a two-stage process within 6000 households from 100 villages in the same provinces as the CBR but outside its catchment area. Using propensity score matching (PSM) and difference in difference analysis, we estimated the differences in accessing services. There were statistically significant differences between participants and controls on the access of available services between the baseline and endline. Using PSM we also found that needs were more often met among CBR participants compared to the controls. Our study indicates that a CBR program may be an effective way to provide services for persons with disabilities even in a conflict context such as Afghanistan. It contributes to addressing the longstanding question whether CBR can actually improve the rehabilitation of persons with disabilities.

## 1. Introduction

Persons with disabilities in Low- and Middle-Income Countries (LMICs) have been marginalised, enduring a level of multipronged difficulties in all spheres of life that no other social group has encountered [1]. Existing evidence points to higher levels of poverty [2,3] and undernutrition [4], food insecurity and poor access [5], a worse health status [6], scarce access to healthcare and rehabilitation services [7,8], less availability of safe water and sanitation [9], a lower access to quality education [10,11,12,13], poor employment opportunities [14,15,16], and a higher risk of social and political exclusion [17,18,19]. As a result, meeting the sustainable development goals for persons with disabilities and the promotion of the “full and equal enjoyment of all human rights and fundamental freedoms” for persons with disabilities envisioned by the 2006 United Nations Convention on the Rights of Persons with Disabilities (UNCRPD) in its article 1, require both the mainstreaming of disability in development initiatives and the mobilisation of extra resources to implement programs that can make a difference in LMICs [20]. 

In LMICs, resources to fight poverty, provide equal opportunities to persons with disabilities, promote their rights, and encourage full participation in society are limited. Community Based Rehabilitation (CBR) was introduced by the World Health Organization WHO) following the Declaration of Alma-Ata in 1978 on primary healthcare coverage as an effective strategy to overcome limited resources and poor access to existing ones for persons with disabilities by offering local rehabilitation initiatives in LMICs. Such initiatives aimed at implementing part or all of the five components of the CBR matrix—health, education, livelihood, and social and empowerment components depending on local needs and priorities [21,22]. The philosophy was based on a bottom-up approach engaging people with disabilities themselves but also their families and communities as well as local governmental and non-governmental providers and establishing partnership with outside actors in sectors not covered by the CBR program. Such a philosophy is also emphasized in the CRPD (article 26). Advocates emphasise the broad scope of the multisectoral approach [23,24,25], its cost effectiveness compared to hospital or rehabilitation centre services [26], its focus on overall wellbeing [27], ownership, and the empowerment of people with disabilities [28,29].

During an international consultation to review CBR program principles organised by the WHO and other Un agencies in 2003 [30], several issues were highlighted: promoting participation and ownership of persons with disabilities [31], developing multisectoral collaboration, involving DPOs, scaling up, and promoting evidence-based practices. All these together with use of local resources, cultural sensitivity, building capacity, and political and financial support have been identified as weaknesses [32,33], although they are essential ingredients for sustainability [21]. 

The evaluation of CBR effectiveness remains scarce in LMICs, making the adoption of evidence-based practices problematic to achieve [34]. Many studies focus on a small sample of CBR participants and provide qualitative information on existing barriers and challenges to participation. Many quantitative studies evaluate the health component of CBR [35], a lot less education, less access to assistive devices [36,37], nutrition [38], immunization [39], livelihoods [40] and social inclusion [41], and almost no empowerment [34]. The existing research does not look usually at various disabilities but instead focus on one condition or type of disability [42]. Few studies focus on service delivery outcomes and the improvement in wellbeing [27,36]. Limited research investigates CBR impact in Low Income Countries. Finally, very few studies use robust impact evaluation methodologies [43].

The access to services can be considered as a key indicator of the performance of the CBR programs. Measuring satisfaction with services as part of access is essential as the literature has shown that persons with disabilities often are confronted to situations where inappropriate services, particularly healthcare services, are offered or they face attitudinal, financial, or physical barriers to access [44,45,46]. Studies have mostly investigated the access to healthcare services for persons with disabilities [7,47,48,49,50]. They rarely assessed the access to services for participants in a CBR program [36]. Similarly, the perceived satisfaction of persons with disabilities with services received is rarely emphasized. 

The present study seeks to contribute to the literature by investigating the effectiveness of a wide range of services delivery—physiotherapy, assistive devices, employment services, education services, and advocacy—for participants of a CBR program in rural Afghanistan with various types of disabilities, including their level of satisfaction with the services received over time compared to treatment as usual through the basic package of health services for a random control group of persons with disabilities. 

## 2. Materials and Methods

### 2.1. Study Location and Population

We conducted a large quasi-randomized field experiment of a CBR program that has been implemented in 13 provinces of northern and eastern Afghanistan since 2004. The CBR is implemented in 13 provinces with four regional project offices based in Ghazni, Jalalabad (Nangharar province), Mazar-e-Sharif (Balkh province), and Takhar (Taloqan province) (See Figure 1). As of 2019, the program covers 48 districts with over 775 staff, 863 (413 female) community volunteers, and 151 (60 composed of female) community-based rehabilitation committees. It provides services to an estimated 2301 persons with disabilities in home-based activities, 1443 children with disabilities in home-based education, and 4324 children with disabilities in schools.

### 2.2. Study Design

We included all new 1680 CBR participants included in the CBR program between July 2012 and December 2013 living in one of the 169 urban or rural catchment areas of the CBR program and identified using a locally developed, tested, and validated questionnaire based on the World Health Organization guidelines for grassroot disability program [51]. Catchment areas are composed of villages or peri-urban neighborhoods called *mahals* where recruited CBR workers live and from where the program expands progressively to include nearby villages until covering extensively each district of the 13 included provinces.

We also included during the same period a group of controls living in 100 randomly selected villages and urban centres of the same provinces but outside of the program catchment areas. At the second stage, 60 households were randomly selected from the social centre (a mosque or an open square) of each village/urban centre yielding a total of 6000 households. From the social centre, a random direction was defined using a spinner, direction in which all households were numbered. The first household was identified using a random number selected between 1 and 60. And all other 59 households were selected using the nearest front door method. A household was defined as a unit that shared a kitchen, an income, and occupied the same flat, house, or compound. Children and adults with disabilities were identified using a locally validated disability-screening tool composed of 34 items for adults (DSQ-34) and 35 for children (DSQ-35) [52]. 

### 2.3. Data Collection

All study participants were interviewed with a locally developed and validated questionnaire that inquired about demographic characteristics, socioeconomic status, access to various CBR services from among the five domains of the CBR matrix (health and rehabilitation, education, livelihood, social inclusion, and empowerment) [21], individual functioning, social participation, and additional needs. The questionnaire was reviewed by experts in Nangarhar province for completeness, content validity, and appropriateness of the questions to the Afghan cultural context. It was then translated into Dari and Pashto and independently back translated into English. Both versions were compared to reconcile discrepancies. We also compared to the initial concepts developed by the researchers the responses provided by a group of 20 CBR participants interviewed in Jalalabad, Nangarhar, and again by another group of 30 persons with disabilities of different age groups, gender, and ethnicity interviewed in Kabul. Respondents were asked the questions as defined by researchers followed by a series of probing questions aiming at capturing their understanding of the questions in light of their own life experience [53]. All study participants were interviewed with the same tool three times between July 2012 and December 2013, between July 2013 and December 2014, and finally between July 2014 and December 2015. Village characteristics information has been collected in January and February 2016.

### 2.4. Study Variables

#### 2.4.1. Outcomes

This study assessed self-reported measures of availability of CBR program services and service recipient satisfaction. We first measured if study participants received any of five types of services—namely physical therapy, assistive devices, employment support, education support, advocacy—and then we asked about quality-of-life improvement of the services received. A dummy variable was created for each set and for each outcome (i.e., received, not received and improved, not improved). For access to services, we considered the use of services as a relevant proxy [54,55] and we asked: “Did you receive any of the following services since you joined the program?”.

Physical Therapy

Physical therapy (PT) is provided in 12 existing physiotherapy centers and at home for both children and adults with disabilities diagnosed with a condition that can be improved through PT. 

Mobility and assistive devices

Mobility and assistive devices are made available free of costs to participants with disabilities with mobility limitations. Prosthesis and orthotics are locally manufactured in four regional workshops by Swedish Committee for Afghanistan and provided to people wounded by ammunitions, landmines, or explosive ordnance. Orthotics have been provided in some cases to children presenting conditions such as poliomyelitis and cerebral palsy. In case of paralysis or other walking limitations, locally built wheelchairs or three wheels bicycles with a system of hand pedals have been used. Similarly, crutches, walking frames, and walking sticks all locally fabricated have been made available for different conditions of mobility problems. Finally, special chairs are fabricated for children with cerebral palsy who need some support while sitting. 

Employment and livelihood support

The employment and livelihood support program has been offering increased services to persons with disabilities since the beginning of the present study. Such services include vocational training (tailoring, carpet weavers, or seamstress), business management training, small loans, in-kind support, establishing saving groups and producer groups engaged in sectors such as dairy, garments, and soap making to launch or support businesses. 

Education support

The CBR program promotes early educational intervention through home-based education for children with disabilities. Following a period of home-based education, children are transferred to a Community Rehabilitation Development Centre, where they are introduced to formal literacy and numeracy for two to three years before transitioning to a mainstream school. 

Advocacy and community mobilization

A variety of activities are included in the generic appellation of “advocacy and community mobilization” as we found through discussion with CBR workers and activity observation during the present study. In practice, advocacy and community mobilizations include a combination of disability sensitization and awareness raising interventions by CBR workers among community members. A major activity is general training and lecturing for families, village elders, and members of the local community called *Shurah* about the rights of people with disabilities aiming at fighting stigma and discrimination in the public sphere. Another important activity is awareness raising about the benefits of inclusion in education among school principals, teachers, and parents of both disabled and non-disabled children to promote open and welcoming schools. Similarly, targeted awareness towards employers to promote inclusion of persons with disabilities in employment is an important dimension of advocacy. Such definition was shared with CBR participants who were asked to acknowledge if any advocacy and community mobilizations activity has been conducted by the CBR program to address problems of discrimination they faced in their community.

Service recipient satisfaction was measured with the following question: “What are the remaining needs you have that the program did not cover?”, from those who received a given service. Participants identified a list of nine possible needs: Education and training, housing, healthcare, income, job opportunity, disability pension, respect from family, respect from community, getting married. 

#### 2.4.2. Exposure Variables

The present study measures exposure to CBR program services delivered to participants compared to a group of controls that received usual services offered by the State either directly or through the subcontracting to Non-Governmental Organizations (NGOs), in particular, the basic package of health services [56]. We assess whether participants and controls found available types of services offered by the CBR program at times of need and if they were satisfied with their experience and found such services useful in addressing their needs. 

#### 2.4.3. Potential Confounders

Several covariates were included in the calculation of the propensity score. We included study participants characteristics at baseline, including gender (female and male), age (continuous), disability type (physical or mobility, sensory, intellectual, mental illness, and multiple or associated disability), cause of disability (birth related, accident related, disease related, or conflict related injury), ethnicity (Pashtun, Tajik, other ethnicities including Azara, Uzbek, and other minorities) education level (no education, some formal education), and level of wealth measured by a welfare index using polychoric principal component analysis based on a list of durable goods (radio, mobile phones, television, pressure cooker, refrigerator, generator, solar panel, sewing machine, bicycle, motorbike, autorickshaw, car, and house) owned by the household [57]. 

We also included community-level variables such as availability of electricity (availability or not in the village), distance to the closest school (continuous), and to the closest healthcare facility (continuous). We did not include age and disability type in the calculation of the propensity score. The age range difference between participants and controls was very narrow. There was potential collinearity between cause and type of disability, therefore we used cause of disability for the calculation of the propensity score and for the adjustment of the model.

### 2.5. Statistical Analysis

Descriptive statistics provide the distribution, and chi-squared tests and *t* tests compare demographic and socioeconomic characteristics, and village characteristics and outcomes of interest (access to and satisfaction for CBR services) between CBR participants and non-participants.

For obvious ethical reasons of NGO commitment to make the CBR program available to all persons with disabilities in its catchment area, we could not randomly select participants in the two arms of the study. In absence of meeting all the conditions for an experiment, and to reduce selection bias, we examined the change overtime on the outcomes of interest comparing CBR participants and non-participants using a difference in difference model after propensity score matching (PSM) [58]. We used PSM to evenly balance the distributions of observable confounding factors across CBR program participants and non-participants for the associations between being in the program and receiving five types of services of interest (i.e., physiotherapy, assistive devices, employment support or education support, and advocacy) as well as satisfaction with services received [59,60]. The method does not impose a specific functional form assumption on how participation in the program effects services access and satisfaction nor does it need any specific identification on the model errors [61].

We estimated a propensity score for each individual representing their probability of receiving the mentioned five types of services, conditional on a set of observed covariates that are recognized to predict both treatment assignment and outcomes of interest (Models 1 to 5). Propensity score matching allows to reduce selection bias linked to the choice of a quasi-experimental study design by controlling for selection on observables that might influence the probability of being in the CBR program intervention or control group [59]. Similarly, we estimated a propensity score for the probability to have remaining needs (models 6 to 14). We included the following covariates—age, gender, disability status and cause, ethnicity, level of education, welfare status, village electricity, distance to school, and healthcare facility—as defined above. These were expected to be balanced between CBR participants and non-participants in each model [62]. We then matched CBR participants and non-participants on the propensity score using the Nearest Neighbor Matching method with a 1:1 ratio, and a 0.25 calliper [58,62,63]. The balancing tests show that propensity score matching using Nearest Neighbor Matching estimator removes most of the bias between the treatment and non-treatment groups: In all analyses, Rubin’s B—which reflects the absolute standardized difference of the means of the propensity score in the treated and control groups—is below 25%, Rubin’s R—the ratio of the treated to control variances of the propensity scores—is within 0.5 and 2 and the percentage bias is below 10% for all covariates (Figure 2 and Figure 3) [64]. We interpreted any remaining difference in the outcomes as the average treatment effect on the treated (ATT), the CBR program. 

In both set of models (access to and satisfaction with services), we controlled in the analysis for personal characteristics that might have an effect on the impact of the programme (e.g., gender, age, ethnicity, type of disability, education level, and assets index) and because all those who are eligible within a catchment area are included in the programme we made the reasonable assumption that participant and control groups have similar characteristics overall [59,65].

In mathematical terms, the average treatment effect (*ATE*) and the average treatment effect on the treated (*ATT*) of the CBR program through its interventions can be estimated using the following formulas:$$ATE=\alpha_{ATE}=E\left(Y^{1}-Y^{0}\right)\;ATT=\alpha_{ATT}=E\left[Y^{1}-Y^{0}|D=1\right]=E\left[Y^{1}\left|D=1]-E[Y^{0}\right|D=1\right],$$
where 1 refers to being in the treatment group and 0 being in the control group. However, the problem is that neither *E*[*Y*^0^||*D* = 1] nor *E*[*Y*^1^||*D* = 0] can be observed, since we did not observe what would have happened to the controls had they received the program, or what would have happened to the treatments had they not received it. We estimated these two counterfactuals by matching each CRB participant with one or more controls that were similar in key characteristics. 

To overcome the possible selection bias of non-random choice of participants and controls and the absence of independence between the effect variable and the treatment variable, we introduced an assumption of conditional independence. We assumed that we observed all the variables (X) that led a person to receive the program. Thus, the estimate for the average treatment effect on the treated (*ATT*) can be obtained by:$$E\left[Y^{1}-\hat{Y^{0}}|D=1\right]=\frac{1}{n_{1}}\underset{iDi=1}{\sum }Y_{1i}-\hat{m_{0}}\left(X_{i}\right)$$
where $\hat{m_{0}}\left(X_{i}\right)$ is the non-parametric estimator of $m_{0}\left(x\right)=E\left(Y|X=x,\;D=0\right).$ Because of the large number of variables (X) on which the CBR program is based, CBR participants and controls are matched on the propensity of being treated given that (X) takes a value of (x):$$p\left(x\right)=\mathrm{Pr}\left(D=1|X=x\right).$$

We estimated these two counterfactuals by matching treatment and control with key characteristics. In both sets of models (access to and satisfaction with services) we assumed the existence of common support, which implies that only CBR participants that have a probability of being treated also found in any of the controls were included in the analysis. Similarly, controls with an extremely low probability of being treated were not included either. This method has the advantage of not requiring any assumption on whether the program has homogeneous or heterogeneous effects on the model errors and by being non-parametric it can be combined with other methods in order to yield more precise impact measures [61].

We combined PSM with the difference in difference (DD) approach in the case of access to services only when two points in time ($Y^{0},Y^{1}$) were considered to account for all unobservable differences that are stable over time, therefore eliminating the risk of selection bias even if some unobservable characteristics that lead to the decision on whether to access the program could not be captured with the variables (X) [66]. We identified the effect of accessing the CBR program by comparing the change in access to different services $E\left[Y_{t+1}^{1}-Y_{t}^{0}|D=1\right]$ of the CBR participants between the period (t) and (t+1) to the counterfactual $E\left[Y_{t+1}^{0}-Y_{t}^{0}|D=1\right]$ they would have experienced in the absence of the program. This counterfactual is approximated by the change in access of services $E\left[Y_{t+1}^{0}-Y_{t}^{0}|D=0\right]$ observed in the control group considering the common trend assumption:$$E\left[Y_{t+1}^{0}-Y_{t}^{0}|D=1\right]=E\left[Y_{t+1}^{0}-Y_{t}^{0}|D=0\right]$$

We used a logistic regression of the likelihood to access the program based on baseline variables (X) for the propensity score estimation. After the propensity score, we estimated the average treatment effect on the treated (*ATT*):(1)$$ATT^{DD-PSM}=\frac{1}{N_{D_{1}}}\underset{i\in D_{1}\cap S}{\sum }\left[\left(Y_{i,t+1}^{1}-Y_{i,t}^{0}\right)-\underset{j\in D_{0}\cap S}{\sum }W_{ij}\left(Y_{j,t+1}^{0}-Y_{j,t}^{0}\right)\right]\;$$
where $D_{1}\;\left(D_{0}\right)$ represents the treatment (control) group, $w_{ij}$ the nearest neighbor matching weights, and S the area of common covariate support. PSM makes the standard DD assumption more plausible by forming statistical twin pairs before performing the DD estimator. PSM-DD allowed for measuring the relative difference in change in outcomes over time between CBR participants and controls and counteract the fact that not all variables that led to the definition of a catchment area could be considered, and thus addresses the bias generated by this limitation. As we only considered unmet remaining needs as a proxy of satisfaction with the program at endline, we used a logistic regression of the likelihood of unmet needs based on the same set of baseline variables (X) for the propensity score estimation. We used R for all analyses.

## 3. Results

### 3.1. Baseline Descriptive Statistics 

Study participants’ demographic, socioeconomic, as well as village characteristics of the sample together with outcomes of interest at baseline are presented in Table 1. Program participants were younger ($\bar{X}$ = 15, SD = 15) on average than the controls ($\bar{X}$ = 31, SD = 21), reflecting the priority given by the NGO to children and youth to address disability at an early age. Males represent approximately two thirds of the sample in both groups. Tajik were slightly overrepresented among CBR participants and minority ethnicities (Hazara, Uzbek, etc.) were underrepresented compared to the controls. Disability at birth represents 60% of CBR participants (30% of control) confirming the priority given by the program to detecting and addressing disability at an early stage. Mobility and physical disability are more prevalent in the CBR than in the controls sample whilst mental illness is almost not represented. Above 80% of both groups had no education. Of note is the fact that material poverty as measured by the assets index was wider spread among controls than CBR participants. This gap in material wealth goes together with a gap in village remoteness. Participants villages were more likely to have electricity and to be located closer to a school or a healthcare facility.

### 3.2. Endline Bivariate Analysis

The participants benefitted overall from the CBR program (see Table 2). Access to physiotherapy, advocacy and awareness were provided to over 80% of the CBR participants who needed it compared to 10% and 14% of the controls. The gap between the CBR participants and the controls was less important for education and employment support as well as for access to assistive devices, but still significant (respectively, over 37, 18, and 44 percentage points). Unmet needs were overwhelmingly and significantly higher among the controls compared to the participants. A gap of 23 percentage points was observed for health, the main domain of intervention of the CBR program. Among adults we observed a significant gap for disability pension (nine percentage points) but not for income as 69% of the CBR participants and 66% of the controls were in need of an income. A similar proportion of the controls and participants above 14 years old (slightly more than 50%) were in need of a job. Interestingly, the remaining unmet needs were lower among the controls than the program participants for education (16 percentage points lower) and marriage (11 percentage points among adults), probably again because the group of controls is older. 

### 3.3. Effect of the CBR Program on Access to Services 

We use PSM-DD to assess the access to services through a CBR program in rural Afghanistan. The findings in Table 3 show a trend of increased access overtime for the treated compared to the controls for all services except for physical therapy. This is probably due to an early offer of physiotherapy services to persons with physical and locomotor difficulties that did not change considerably during the time spent in the program. 

We observe overall a significant better access to services for the CBR participants compared to the controls. Participating in the CBR program is associated with, respectively, a 94.9, 17, 31.6, 3.7, and 30 percentage-point higher probability of accessing physical therapy, assistive technology, employment, education and advocacy, and community support compared to the controls, net of individual level covariates. Among the control variables, females were significantly less likely to receive assistive technology and support to access to school. Pashtun were significantly more likely to benefit from support to education than Tajik and from physical therapy services than minority ethnicities. An increasing age of school going age children between six and 17 years old predicted a lower probability of support to education, confirming the effort made by the program to support school inclusion early on in life. People disabled at birth were more likely to benefit from employment and educational support than people disabled from any other cause. One exception was employment support for those disabled due to war, but the difference was not significant. In contrast, people disabled due to war were more likely to benefit from both physical therapy and assistive technology services. This is explained by the fact that many of them had contractures or loss of limbs resulting from a war wound necessitating either or both of the services. Educated adults were more likely to receive employment support indicating that the CBR program helped more of those individuals more likely to be recruited. Physical therapy and assistive technology services were more accessible to the richest individuals. Advocacy and community mobilization activities were more likely to take place in remote villages without electricity but not educational support services. Finally, the distance to school reduced the likelihood of all services except educational support services, while the distance to the closest healthcare facility did not influence the provision of any service.

### 3.4. Unmet Needs by the CBR Program 

Table 4, Table 5 and Table 6 present the results of the logistic regression models about the difference in unmet needs comparing the participants with disabilities in a CBR program to the controls with disabilities in rural and peri urban Afghanistan with covariates. Unadjusted models showed a positive effect of the CBR program in significantly reducing all needs except the need for education for those 6–18, jobs for those between the ages of 16 and 60, income for those 18 and older, respect from family, and marriage for participants compared to the controls. The effect became significant for income when controlling for our covariates and remained unchanged in its amplitude for healthcare and housing. By contrast, the effect was amplified in the adjusted model for, respectively, 1.54 (95%CI 1.11–2.17), 2.3 (1.72–2.94), and 1.59 (1.20–2.13) times less likely to have unmet needs of family and community respect as well as unmet marriage needs. These findings reflect the importance given by the CBR workers to sensitization and awareness of rights of persons with disabilities. 

Among the control variables, being a female significantly increased the risk (1.28 95%CI 1.01–1.62) of having unmet health needs and male while being a Pashto significantly increased the risk of having unmet educational and pension needs. Other minority ethnicities were more likely to have a remaining need to find a life partner than Pashto. Not surprisingly, unmet needs for income and pension increased with age, while educational and social (respect and marriage) needs were less likely to be unmet with aging. Conversely, people with intellectual and associated disabilities compared to persons with a physical disability as well as those disabled at birth compared to those disabled from a known cause such as war violence, a disease, or an accident, were more likely to lack family and community respect, a noticeable indication of the stigma associated to certain types and causes of disability. Uneducated people were also more at risk of unmet needs for respect. Finally, the poorest were more likely to have unmet needs in all domains. 

## 4. Discussion

Our study offers several contributions to the literature on the impact of CBR programs in LMICs. First, we provide insights into the effectiveness of in the delivery of health and rehabilitation, employment support, access to school and advocacy, and community mobilization services of a CBR program in Afghanistan using a quasi-experiment approach. Our results revealed an overall significant impact of the CBR program on the access to physical therapy, assistive technology, employment, education and advocacy, and community mobilization, indicating that CBR programs can bring closer needed services to persons with disabilities in LMICs. Second, we established participants’ satisfaction with the program coverage by investigating the level of uncovered remaining needs—health, education, housing, employment, income, pension, family and community respect, and marriage—that belong to various components of the WHO CBR matrix [21]. Third, our large-scale rigorous quasi experiment following a large sample of persons with disabilities for three years informs the limited literature examining the impact of the CBR program in Low- and Middle-Income Countries, relying mostly on observational or qualitative studies of (LMICs) [34,67].

The CBR program had a differential impact in terms of access to services. The highest positive effect was on the access to physical therapy services. After 3 years on average in the program, the estimated marginal effect is a considerable 105.8%. The large gap between treated and controls is the result of a long-standing offering of physical therapy services but also assistive technology devices, both at the origin of the CBR program in Afghanistan, a country where decades of conflict have generated a large number of people physically disabled by violence and in need of such services [68]. These services are crucial in increasing autonomy in activities of daily living for persons with mobility impairment, which results in lessening the burden of assistance for caregivers that otherwise rests on them (55): The significant and large effect on the probability of being employed among CBR participants compared to the controls is of note considering that people with disabilities are often excluded from employment opportunities [1], including in Afghanistan [69], and that CBR programs are rarely able to make a significant advance in this specific domain [36], despite its central role in fostering social inclusion, autonomy, self-esteem, and the overall quality of life for persons with disabilities [70]. Likewise, the positive impact of advocacy campaigns and community support are instrumental in modifying existing mindsets, lowering attitudinal barriers, as well as in reducing stigma towards persons with disabilities, a major step towards improving inclusion in multiple spheres of life such as employment, education, healthcare access, and social life [47,71,72,73]. 

The improvement in access to services is not uniform across the sample of participants. In particular, the gap in the benefit of assistive technology between males and females is partially explained by the lower number of females who lost limbs or were injured due to the conflict. The program put an emphasis on intervening earlier in life and/or at the onset of disability to try and minimize the negative impact of the impairment explaining an early intervention for education. The program also aims at supporting those who are most stigmatized. This probably explains why people disabled due to the conflict were less likely to receive employment support compared to those disabled at birth. Literature has shown that in Afghanistan, persons who were disabled due to conflict are highly recognized in Afghan society and often benefit from socioeconomic advantages, in particular in terms of job opportunities. It is probably why the recipients of physical therapy and assistive technology are more likely to be found among the richest 20%. Conversely, persons who were disabled at birth or from an unknown cause, are stigmatized against and considered disabled as the consequence of a curse of God, spirits, jinn, fate (*kismet*), or black magic. The latter are designed by the use of a derogative term ‘*mayub*’ in Dari and Pashto, the two major languages in use [74]. To fight ingrained cultural stereotypes within the context of Afghanistan, the CBR program has been advocating for the rights to access services for persons disabled at birth, particularly in remote areas as a way to promote their overall wellbeing. The stigma of disability has been shown to limit access to healthcare [7,75] and access to school [76,77], reducing opportunities to make friends and build a family life [78], resulting in mental distress [79]. 

Our study also established that the CBR program significantly reduced multiple unmet basic needs (healthcare, education, employment, housing, and social inclusion) of program participants compared to the controls. We did not find such a gap in reducing unmet needs for disability pension. A possible explanation for the absence of a significant reduction in the need for a pension is the small amount of the existing disability pension scheme unlikely to cover basic capabilities, combined with the eligibility rules that favor persons wounded or killed at war (‘martyrs’), excluding de facto many persons with disabilities from its benefit. 

But the otherwise reduction in unmet needs is extremely important considering that a growing body of literature has demonstrated a higher level of deprivation of people with disabilities not only in terms of income and material wealth but also in terms of basic needs, compared to the rest of the population in LMICs [80,81,82,83,84]. Going a step further, Sen (2009) and followers have argued that persons with disabilities are actually deprived of capabilities, which are opportunities to live the life they value [85]. For Sen, poverty is understood as deprivation: “deprivation of basic capabilities rather than merely as lowness of incomes” [86] p. 87. Such basic capabilities include education, nutrition, healthcare, employment, and housing [87]. Overall, persons with disabilities have fewer opportunities and less agency resulting in a loss of wellbeing and in the deprivation of basic capabilities [88]. For instance, drastically reducing healthcare and rehabilitation needs is instrumental in promoting the overall wellbeing of persons with disabilities in a country where a recent study has shown that the perceived access to the mainstream healthcare system for persons with disabilities has not improved between 2004 and 2017 [7]. Similarly, we found a differential reduction in housing, employment, and income that also contribute to improved wellbeing. Adequate and affordable housing is part of the livelihood’s component of the CBR guidelines [21] and more generally of the sustainable development goals (SDG 11). It has been argued that persons with disabilities face additional difficulty associated with discrimination and possibly extra cost [89] to secure accessible housing in a population already facing financial hardship due to limited access to employment [1]. Trani et al. (2016) have shown that persons with disabilities in Afghanistan face worse living conditions than non-disabled people, including living more often in crowded spaces [90]. The same study demonstrated that Afghans with disabilities were more often unemployed and deprived of assets than the rest of the population. Reducing overall unmet basic needs contributed to the considerably improved overall wellbeing of CBR participants by demonstrating that opportunities to do and be what they value were more often offered to them through the CBR program. 

The impact of the CBR program on the level of respect from family and the community as well as access to marriage is an important finding with far-reaching consequences. First, building family and community respect has proven effective in reducing internalized stigma, low self-esteem [91], and mental distress [79]. Social inclusion and community participation are essential capabilities that are instrumental in fostering the wellbeing of persons with disabilities. Second, challenging communities’ attitudes and fostering persons with disabilities’ social participation is a central mission of the CBR program [71]. Yet, there is scant evidence that CBR programs are effectively fighting stigma [32,33]. Third, changing attitudes towards persons with disabilities has been shown to be an extremely challenging and slow process [92], and our achievement after three years of program is therefore very encouraging. 

There is notable variation in unmet needs according to people with disability characteristics. Poorer disabled people remained more likely to have unmet needs than wealthier disabled people. This finding is supported by the available evidence showing that the material deprivation of persons with disabilities is associated with the deprivation of other basic capabilities such as employment, healthcare, education, and social and political participation [19,93,94] among but also with low self-esteem, the deterioration of social networks, and the loss of meaning of life for people with disabilities [95]. Women were more likely to have remaining healthcare and housing needs. They expressed less often the need for employment and income than men. This is because of traditional norms under the *Pashtunwali* code that determines that men are the breadwinners while women should not participate in social and economic life and access public areas, including healthcare facilities, unless accompanied by a *mahram*, a male close relative, but instead fulfill a role of house wife and mother [96]. It is of note that women expressed a higher need for family and community respect, probably the result of both a diminished status of women that are under the authority of men, but also in many cases the result of existing domestic violence associated with the conflict source of economic struggle that creates frustration and mental distress in men unable to meet their social obligations as breadwinners [97,98]. Likewise, older men with disabilities are more often likely to be the head of the family, which explains a less unmet need of respect but conversely a more unmet need in terms of income and pension, as they are expected to provide livelihoods for family members. This situation of gender inequality is bound to persist or even increase under the new Taliban regime. Persons with intellectual or associated disabilities, disabled at birth, or from an unknown cause were also more likely to feel not respected and in need of being able to get married compared to persons with a physical disability. This finding confirms the entrenched stigma associated in Afghanistan with intellectual and associated disability or disability without a clear cause that translates into pervasive discrimination [74].

Our study presents some limitations. First, the quasi-experiment approach we had to adopt does not ensure random assignment to either the treatment or the control arm, presenting a risk of selection bias. Elderly people and people with associated disabilities were significantly less likely to participate in the program (53). Similarly, the program coverage area included villages that seemed less remote than the control villages when considering better electricity coverage and less distance to the closest school and healthcare facility. To reduce the risk of selection on non-observable variables, we included multiple control variables to calculate the propensity score. We also used a difference in difference model to increase the likelihood of unbiased estimates of the CBR program effect. Baseline information shows that CBR participants started worse off than their counterparts, reducing the chance that observed impacts were due to initial differences favoring CBR participants. Similarly, services sometimes started immediately, in particular for physical therapy, showing a high proportion of CBR participants already receiving services at baseline. Providing treatment and services came first, the study came second. Therefore, we observed a vast majority of people with mobility limitations who already started receiving physiotherapy from the program at the time of their first interview. There is also a risk of information bias as data collection was carried out by CBR workers for security reasons. This might have encouraged a social desirability bias as CBR workers might have been tempted to show a positive impact of their program. We conducted a close supervision of data collection, random re-interviews, and multiple consistency checks to reduce the risk of bias. 

## 5. Conclusions

This empirical study has shown that a CBR program could contribute to improve the livelihood of persons with disabilities, lessen the costs of living with a disability, promote their participation in family and community life, escape stigma and prejudice, and reduce the level of unmet needs by effectively ensuring access to various CBR services from among the five domains of the CBR matrix (health, education, livelihood, social inclusion, and empowerment) in Afghanistan, a country characterized by over four decades of turmoil. A key message arising from this study is that CBR programs can effectively improve the social and economic life of persons with disabilities and their families, even in a crisis context. Consequently, CBR programs can also improve the mental wellbeing of persons with disabilities by reducing the impact of the multiple daily stressors they endure [79,99]. These are a major finding considering that few studies have evaluated the impact of a CBR program in an LMIC [27,32,36,92], none in a conflict context, and none following a large group of study participants for a period of three years with three waves of interviews. Future research is warranted to further define the tools and mechanisms for the standardization of CBR workers’ practice. Tools would include a new instrument used for monitoring the recruitment process of new participants and their progress. The standardization of practices would cover criteria for inclusion in, and the implementation of, home-based therapy, advocacy, and economic support (allocation of loans and vocational trainings). Another field of future research consists of assessing the impact of the continuous capacity building of CBR staff. Finally, future research could examine the greater participation of end beneficiaries and their families, as well as the organization of persons with disabilities in the definition of services and activities provided by the CBR program. Since the present study took place and to date, and despite the recent socio-political changes, the Swedish Committee for Afghanistan has continued to provide services to Afghan with disabilities and expand the home based disability program, covering more villages in the same 13 provinces [100].

## Figures and Tables

**Figure 1 ijerph-19-06341-f001:**
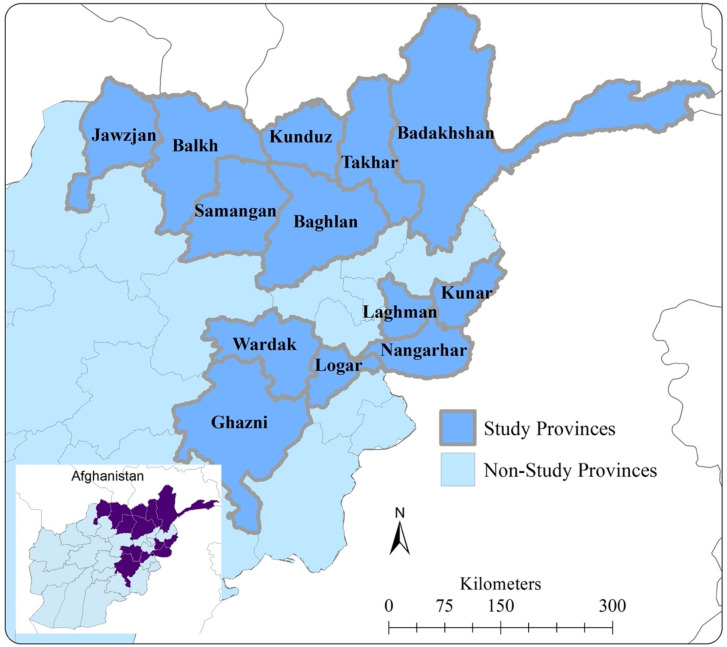
Provinces of intervention of the Swedish Committee for Afghanistan home based disability program and areas of study.

**Figure 2 ijerph-19-06341-f002:**
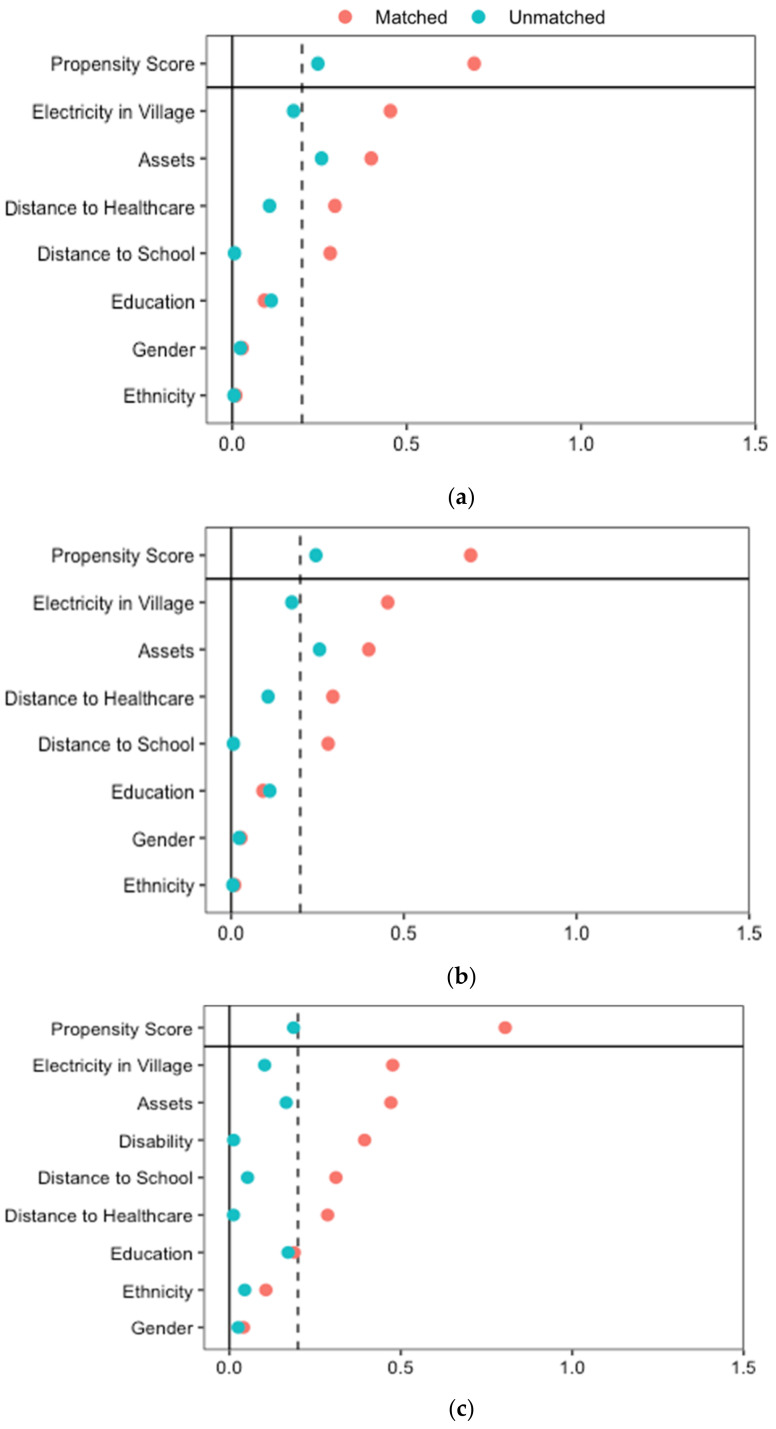
Balance results of the propensity-score matching for the sample for five models of access to services. Model 1: Physiotherapy, Model 2: Assistive devices, Model 3: Employment, Model 4. Education, and Model 5. Advocacy and community mobilization. (**a**) Model 1: Comparative access to physiotherapy; (**b**) Model 2: Comparative access to assistive devices; (**c**) Model 3: Comparative access to employment and livelihood support; (**d**) Model 4: Comparative access to education support; (**e**) Model 5: Comparative access to advocacy and community mobilization.

**Figure 3 ijerph-19-06341-f003:**
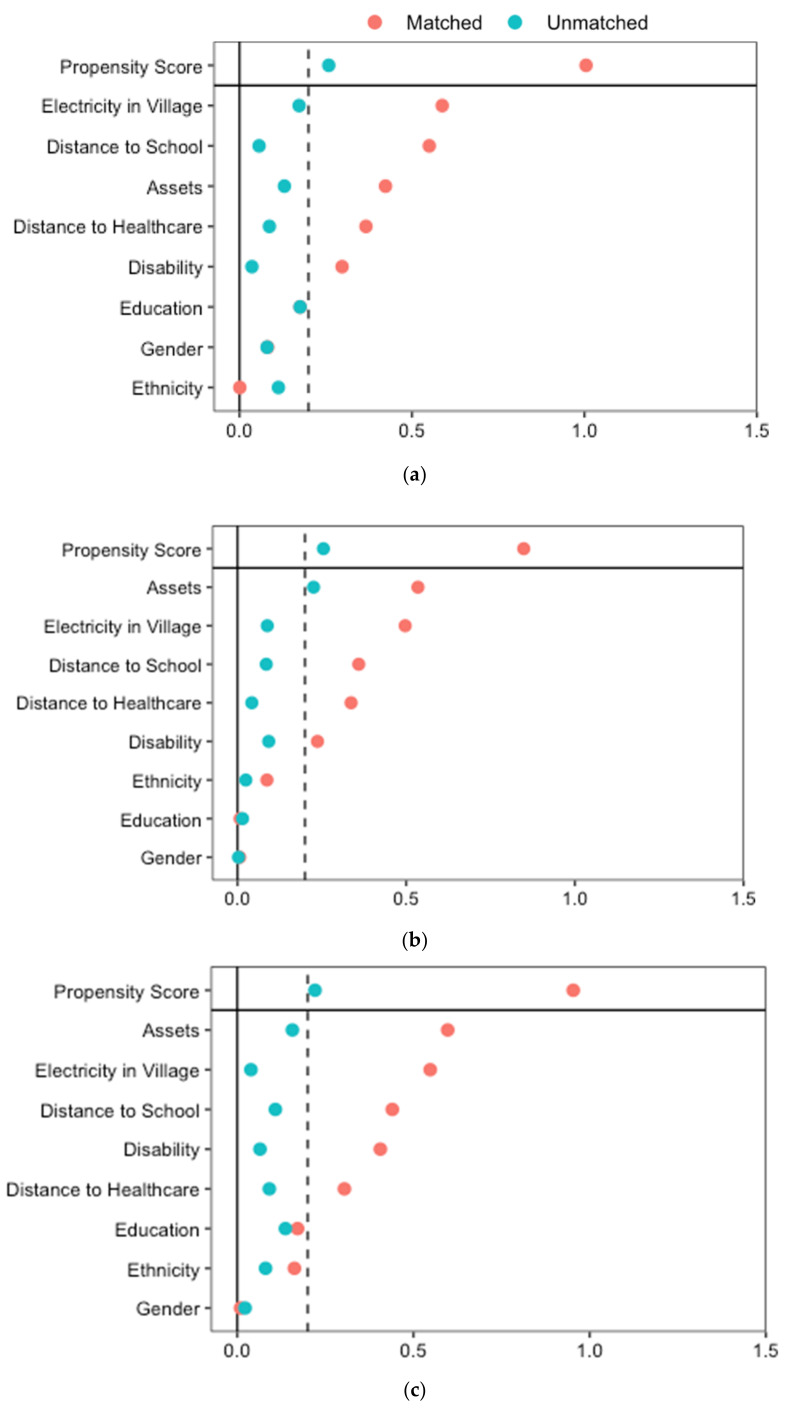
Balance results of the propensity-score matching for the sample for unmet needs. Model 6: Education, Model 7: Health, Model 8: Employment, Model 9: Housing, Model 10: Income, Model 11: Pension, Model 12: Family respect, Model 13: Community Respect, Model 14: Marriage. (**a**) Model 6: Comparative unmet education needs; (**b**) Model 7: Comparative unmet health needs; (**c**) Model 8: Comparative unmet employment needs; (**d**) Model 9: Comparative unmet housing needs; (**e**) Model 10: Comparative unmet income needs; (**f**) Model 11: Comparative unmet pension needs; (**g**) Model 12: Comparative unmet family respect needs; (**h**) Model 13: Comparative unmet community respect needs; (**i**) Model 14: Comparative unmet marriage needs.

**Table 1 ijerph-19-06341-t001:** Baseline study sample characteristics.

	Control (*n* = 1133)	Treatment (*n* = 1861)	Overall (*n* = 2994)
Gender			
Male	701 (62%)	1159 (62%)	1860 (62%)
Female	431 (38%)	702 (38%)	1133 (38%)
Age (years)			
Mean (SD)	31 (21)	15 (15)	21 (19)
Ethnicity			
Pashtun	339 (30%)	669 (36%)	1008 (34%)
Tajik	345 (30%)	749 (40%)	1094 (37%)
Other	274 (24%)	429 (23%)	703 (23%)
Disability			
Physical	567 (50%)	1307 (70%)	1874 (63%)
Sensory	178 (16%)	230 (12%)	408 (14%)
Intellectual	70 (6%)	176 (9%)	246 (8%)
Mental Illness	45 (4%)	7 (0%)	52 (2%)
Multiple	98 (9%)	130 (7%)	228 (8%)
Disability Cause			
Birth	338 (30%)	1118 (60%)	1456 (49%)
Accident	164 (14%)	242 (13%)	406 (14%)
Disease	296 (26%)	336 (18%)	632 (21%)
Conflict-related Injury	117 (10%)	94 (5%)	211 (7%)
Other	217 (19%)	71 (4%)	288 (10%)
Education			
No formal education	780 (69%)	1547 (83%)	2327 (78%)
Some formal education	177 (16%)	295 (16%)	472 (16%)
Assets			
20% poorest	341 (30%)	231 (12%)	572 (19%)
20–80%	532 (47%)	1183 (64%)	1715 (57%)
20% richest	139 (12%)	433 (23%)	572 (19%)
Electricity in Village			
No	264 (23%)	184 (10%)	448 (15%)
Yes	868 (77%)	1677 (90%)	2545 (85%)
Time to Reach School (minutes)			
Mean (SD)	19 (12)	17 (10)	18 (11)
Time to Reach Healthcare (minutes)			
Mean (SD)	30 (24)	25 (18)	27 (21)

**Table 2 ijerph-19-06341-t002:** Bivariate analysis.

	Control *n* (%)	Treatment *n* (%)	*p* Value
Access to services			
Physical Therapy	100 (8.83)	1514 (81.35)	0 < 0.001
Assistive Service	102 (9.01)	1001 (53.80)	0 < 0.001
Employment Service	212 (41.57)	253 (48.47)	0.026
Education Service	22 (8.87)	391 (6306)	0 < 0.001
Advocacy Service	158 (13.96)	1578 (84.79)	0 < 0.001
Unmet needs			
Healthcare	496 (56.56)	444 (33.66)	0 < 0.001
Education (6–17)	186 (75)	459 (74.51)	0.882
Housing	205 (23.38)	209 (15.88)	0 < 0.001
Employment (15–61)	298 (58.43)	269 (51.53)	0.026
Pension (>17)	492 (56.1)	545 (41.44)	0 < 0.001
Income (>17)	412 (68.9)	331 (65.81)	0.275
Family respect	160 (18.24)	191 (14.48)	0.018
Community respect	388 (44.24)	419 (31.77)	0 < 0.001
Marriage	209 (31.67)	266 (43.89)	0 < 0.001

**Table 3 ijerph-19-06341-t003:** Results of linear probability models for access to services.

	Physical Therapy	Assistive Technology	Employment	Education	Advocacy
	β	β	β	β	β
Endline (ref.: Baseline)	0.087 ***	0.105 ***	0.043	0.027	0.105 ***
	(0.014)	(0.024)	(0.035)	(0.024)	(0.017)
Treatment (ref.: Control)	0.949 ***	0.170 **	0.316 ***	0.037	0.300 ***
	(0.032)	(0.054)	(0.079)	(0.055)	(0.038)
Female (ref.: Male)	0.021	−0.068 ***	0.011	−0.055 **	0.012
	(0.011)	(0.019)	(0.026)	(0.020)	(0.012)
Tajik (ref.: Pashtun)	−0.022	−0.012	0.002	−0.046 *	−0.005
	(0.012)	(0.020)	(0.030)	(0.021)	(0.014)
Other ethnicity (ref.: Pashtun)	−0.042 **	0.008	−0.008	−0.030	0.024
	(0.013)	(0.022)	(0.033)	(0.022)	(0.016)
Age	−0.001	0.000	−0.005	−0.003 ***	−0.001
	(0.000)	(0.001)	(0.004)	(0.001)	(0.000)
Accident (ref.: Birth)	0.005	−0.022	−0.105 *	−0.062 *	−0.002
	(0.016)	(0.027)	(0.042)	(0.027)	(0.019)
Disease (ref.: Birth)	0.026	−0.022	−0.036	−0.076 **	−0.003
	(0.015)	(0.026)	(0.035)	(0.026)	(0.017)
War related (ref.: Birth)	0.037	0.087 **	−0.196 *	0.033	0.020
	(0.019)	(0.032)	(0.099)	(0.031)	(0.025)
Other cause (ref.: Birth)	−0.013	−0.108 *	−0.105	−0.068	−0.039
	(0.032)	(0.053)	(0.073)	(0.044)	(0.033)
Education (ref.: No education)	0.011	0.035	0.167 ***	0.045 *	0.029
	(0.013)	(0.022)	(0.030)	(0.022)	(0.016)
Assets 20–80% (ref.: 20% poorest)	−0.001	0.002	−0.036	0.028	−0.003
	(0.013)	(0.022)	(0.033)	(0.023)	(0.015)
Assets 20% richest (ref.: 20% poorest)	0.028	0.076 **	−0.019	0.001	0.023
	(0.016)	(0.027)	(0.042)	(0.029)	(0.019)
Electricity in village (ref.: No electricity)	−0.004	0.063 **	0.033	0.059 *	−0.081 ***
	(0.014)	(0.023)	(0.033)	(0.025)	(0.016)
Time to reach to school	−0.002 ***	−0.001	−0.000	0.003 ***	−0.003 ***
	(0.000)	(0.001)	(0.001)	(0.001)	(0.001)
Time to reach healthcare facility	0.000	−0.000	−0.001	0.001	−0.000
	(0.000)	(0.000)	(0.001)	(0.001)	(0.000)
Wave time × Treatment	−0.052 *	0.170 ***	0.124 *	0.184 ***	0.203 ***
	(0.020)	(0.034)	(0.050)	(0.035)	(0.024)
N. obs.	2164	2164	932	1656	3620
R squared	0.780	0.282	0.345	0.256	0.466
F statistic	448.707	49.633	28.295	33.114	185.046
*p* value	0.000	0.000	0.000	0.000	0.000

Note: The βs refer to regression coefficients estimated by linear probability models. Level: *** *p* < 0.001; ** *p* < 0.01; * *p* < 0.05. Robust standard errors in parentheses; No multicollinearity between the independent variables was detected.

**Table 4 ijerph-19-06341-t004:** Results of logistic models on health, education, and housing needs.

	Health			Education			Housing		
Variable	OR ^1^	95% CI ^1^	*p*-Value	OR ^1^	95% CI ^1^	*p*-Value	OR ^1^	95% CI ^1^	*p*-Value
Group									
Control	Reference			Reference			Reference		
Treatment	0.51	0.40, 0.66	<0.001	1.02	0.66, 1.58	>0.9	0.69	0.51, 0.93	0.016
Gender									
Male									
Female	1.28	1.01, 1.62	0.045	0.75	0.48, 1.16	0.2	1.16	0.87, 1.54	0.3
Ethnicity									
Pashtun	Reference			Reference			Reference		
Tajik	1.10	0.85, 1.43	0.5	0.70	0.42, 1.17	0.2	1.12	0.81, 1.54	0.5
Other	0.79	0.59, 1.07	0.13	0.57	0.33, 0.99	0.044	1.06	0.74, 1.50	0.8
Age (years)	1.00	0.99, 1.01	0.9	0.92	0.87, 0.97	0.004	1.00	0.99, 1.01	0.5
Disability									
Physical	Reference			Reference			Reference		
Sensory	0.71	0.51, 0.98	0.037	1.41	0.81, 2.47	0.2	0.73	0.49, 1.09	0.13
Intellectual	1.20	0.80, 1.79	0.4	1.71	0.91, 3.28	0.10	0.79	0.46, 1.30	0.4
Mental Illness	1.81	0.87, 4.02	0.12	0.65	0.28–1.46	0.3	0.55	0.20, 1.31	0.2
Associated	1.43	0.94, 2.19	0.10	2.24	1.09, 4.92	0.035	0.75	0.43, 1.23	0.3
Disability Cause									
Birth	Reference			Reference			Reference		
Accident	0.75	0.51, 1.10	0.14	1.00	0.49, 2.10	>0.9	1.05	0.66, 1.65	0.8
Disease	0.90	0.65, 1.26	0.6	0.90	0.51, 1.64	0.7	1.37	0.92, 2.03	0.12
War related	0.61	0.38, 0.98	0.041	0.62	0.14, 2.69	0.5	1.51	0.88, 2.55	0.13
Other	1.05	0.54, 2.04	0.9	0.25	0.06, 0.86	0.034	2.08	1.03, 4.06	0.035
Education									
No formal education	Reference			Reference			Reference		
Formal Education	0.54	0.39, 0.74	<0.001	0.40	0.24, 0.65	<0.001	1.21	0.84, 1.72	0.3
Assets									
20% Poorest	Reference			Reference			Reference		
20–40%	0.75	0.53, 1.05	0.093	1.89	0.97, 3.72	0.061	0.72	0.49, 1.06	0.10
40–60%	0.61	0.42, 0.86	0.005	0.99	0.51, 1.90	>0.9	0.58	0.38, 0.88	0.010
60–80%	0.44	0.30, 0.64	<0.001	0.96	0.49, 1.89	>0.9	0.55	0.35, 0.85	0.008
20% Richest	0.57	0.40, 0.82	0.002	1.51	0.75, 3.04	0.2	0.48	0.31, 0.74	0.001
Electricity in Village									
No Electricity	Reference			Reference			Reference		
Electricity	1.22	0.90, 1.65	0.2	0.73	0.42, 1.24	0.2	1.09	0.76, 1.58	0.7
Time to School (minutes)	1.01	1.00, 1.03	0.013	1.00	0.98, 1.02	>0.9	1.00	0.99, 1.01	0.7
Time to Healthcare (minutes)	1.02	1.01, 1.02	<0.001	1.00	0.99, 1.01	0.7	1.00	1.00, 1.01	0.2

^1^ Note: OR = Odds Ratio, CI = Confidence Interval.

**Table 5 ijerph-19-06341-t005:** Results of logistic models on employment, income, and pension needs.

	Employment			Income			Pension		
Variable	OR ^1^	95% CI ^1^	*p*-Value	OR ^1^	95% CI ^1^	*p*-Value	OR ^1^	95% CI ^1^	*p*-Value
Treatment									
Control	—	—		—	—		—	—	
Treatment	0.74	0.54, 1.01	0.062	0.55	0.38, 0.80	0.002	0.62	0.44, 0.88	0.007
Gender									
Male	—	—		—	—		—	—	
Female	0.82	0.59, 1.15	0.2	0.59	0.40, 0.86	0.006	0.74	0.51, 1.06	0.10
Ethnicity									
Pashtun	—	—		—	—		—	—	
Tajik	1.67	1.17, 2.41	0.005	1.70	1.14, 2.55	0.009	0.68	0.46, 1.00	0.048
Other	1.10	0.75, 1.62	0.6	1.25	0.81, 1.93	0.3	0.36	0.23, 0.54	<0.001
Age (years)	0.97	0.96, 0.98	<0.001	0.97	0.96, 0.98	<0.001	1.00	0.99, 1.01	0.6
Disability									
Physical	—	—		—	—		—	—	
Sensory	1.06	0.66, 1.71	0.8	1.39	0.81, 2.44	0.2	1.09	0.65, 1.84	0.7
Intellectual	1.22	0.58, 2.66	0.6	1.23	0.44, 3.82	0.7	1.93	0.71, 5.87	0.2
Mental Illness	1.12	0.43, 3.04	0.8	0.68	0.25, 1.99	0.5	0.59	0.21, 1.60	0.3
Multiple	0.69	0.36, 1.33	0.3	0.58	0.27, 1.27	0.2	0.90	0.41, 2.00	0.8
Disability Cause									
Birth	—	—		—	—		—	—	
Accident	1.02	0.64, 1.65	>0.9	1.71	0.97, 3.03	0.062	1.35	0.78, 2.32	0.3
Disease	1.20	0.76, 1.89	0.4	1.25	0.74, 2.12	0.4	0.96	0.58, 1.60	0.9
Conflict-related	1.09	0.63, 1.89	0.8	1.00	0.55, 1.82	>0.9	0.63	0.35, 1.11	0.11
Other	0.64	0.25, 1.61	0.3	1.21	0.49, 3.08	0.7	0.54	0.22, 1.29	0.2
Education									
No formal education	—	—		—	—		—	—	
Formal Education	0.77	0.53, 1.12	0.2	0.96	0.62, 1.50	0.9	0.91	0.60, 1.38	0.7
Assets									
120% Poorest	—	—		—	—		—	—	
20–40%	0.76	0.47, 1.24	0.3	0.76	0.42, 1.35	0.3	0.90	0.52, 1.55	0.7
40–60%	1.18	0.71, 1.94	0.5	0.90	0.50, 1.63	0.7	1.27	0.73, 2.21	0.4
60–80%	0.82	0.49, 1.37	0.5	1.00	0.54, 1.82	>0.9	1.01	0.57, 1.76	>0.9
20% Richest	0.79	0.48, 1.29	0.3	0.56	0.31, 0.99	0.048	0.67	0.38, 1.16	0.2
Electricity in Village									
No Electricity	—	—		—	—		—	—	
Electricity	0.71	0.44, 1.13	0.2	1.01	0.59, 1.70	>0.9	1.20	0.72, 1.98	0.5
Time to School (minutes)	1.00	0.99, 1.02	0.7	1.01	0.99, 1.03	0.2	1.01	0.99, 1.03	0.4
Time to Healthcare (minutes)	1.01	1.00, 1.01	0.2	1.01	1.00, 1.02	0.2	1.01	1.00, 1.02	0.044

^1^ OR = Odds Ratio, CI = Confidence Interval.

**Table 6 ijerph-19-06341-t006:** Results of logistic models on family or community respect and marriage needs.

	Family Respect			Community Respect			Marriage		
Characteristics	OR ^1^	95% CI ^1^	*p*-Value	OR ^1^	95% CI ^1^	*p*-Value	OR ^1^	95% CI ^1^	*p*-Value
Group									
Control	Reference			Reference			Reference		
Treatment	0.65	0.46, 0.90	0.011	0.44	0.34, 0.58	<0.001	0.63	0.47, 0.83	0.001
Gender									
Male	Reference			Reference			Reference		
Female	1.28	0.94, 1.75	0.12	1.02	0.80, 1.31	0.9	0.98	0.74, 1.28	0.9
Ethnicity									
Pashtun	Reference			Reference			Reference		
Tajik	1.18	0.83, 1.67	0.4	1.46	1.11, 1.92	0.007	1.18	0.87, 1.59	0.3
Other	0.89	0.59, 1.33	0.6	1.27	0.94, 1.73	0.12	1.55	1.11, 2.15	0.009
Age (years)	0.98	0.97, 0.99	0.006	0.98	0.98, 0.99	<0.001	0.98	0.97, 0.99	<0.001
Disability									
Physical	Reference			Reference			Reference		
Sensory	0.82	0.51, 1.28	0.4	1.03	0.73, 1.43	0.9	1.16	0.81, 1.66	0.4
Intellectual	2.08	1.30, 3.29	0.002	2.66	1.75, 4.07	<0.001	1.11	0.71, 1.72	0.6
Mental Illness	1.97	0.82, 4.42	0.11	1.90	0.91, 3.99	0.087	1.29	0.54, 2.82	0.5
Associated	2.33	1.44, 3.70	<0.001	2.05	1.35, 3.12	<0.001	1.10	0.69, 1.72	0.7
Disability Cause									
Birth	Reference			Reference			Reference		
Accident	0.51	0.28, 0.90	0.026	0.73	0.49, 1.09	0.13	0.70	0.45, 1.08	0.11
Disease	0.90	0.57, 1.39	0.6	0.66	0.46, 0.93	0.019	0.84	0.58, 1.22	0.4
Conflict-related	0.41	0.16, 0.91	0.041	0.46	0.26, 0.77	0.004	0.79	0.45, 1.35	0.4
Other cause	0.77	0.27, 1.88	0.6	1.00	0.49, 2.03	>0.9	1.86	0.88, 3.86	0.10
Education									
No formal education	Reference			Reference			Reference		
Formal Education	0.37	0.20, 0.63	<0.001	0.31	0.22, 0.45	<0.001	2.88	2.10, 3.95	<0.001
Assets									
20% Poorest	Reference			Reference			Reference		
20–40%	0.68	0.44, 1.05	0.082	0.88	0.62, 1.25	0.5	0.65	0.44, 0.96	0.032
40–60%	0.51	0.31, 0.83	0.007	0.69	0.47, 1.00	0.049	0.96	0.64, 1.42	0.8
60–80%	0.56	0.34, 0.92	0.023	0.65	0.44, 0.96	0.033	0.97	0.64, 1.46	0.9
20% Richest	0.70	0.44, 1.11	0.13	0.91	0.62, 1.31	0.6	0.89	0.59, 1.33	0.6
Electricity in Village									
No Electricity	Reference			Reference			Reference		
Electricity	0.90	0.61, 1.36	0.6	1.00	0.73, 1.37	>0.9	0.77	0.55, 1.07	0.12
Time to School (minutes)	1.02	1.00, 1.03	0.027	1.00	0.99, 1.01	0.7	1.00	0.99, 1.01	>0.9
Time to Healthcare (minutes)	1.00	0.99, 1.01	0.5	1.00	1.00, 1.01	0.5	1.00	0.99, 1.00	0.6

^1^ Note: OR = Odds Ratio, CI = Confidence Interval.

## Data Availability

The data will be made available in the DRYAD repository at https://datadryad.org/stash (accessed on 28 April 2022) with investigator support, after approval of a proposal, with a signed data access agreement.

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
