# Peer review of "Access to Services from Persons with Disabilities in Afghanistan: Is Community Based Rehabilitation Making a Difference?"

_ijerph, 2022, doi:10.3390/ijerph19106341_

Round 1

Reviewer 1 Report

The authors analyzed the access to communty based rehabilitation services from persons with disabilities in Afghanistan. It is an interesting paper and smooth read. I really enjoyed the introduction, results and discussion sections. I do not have any major concern. Few minor concerns and suggestion are:

Abstract: PSM should be used on Line 23 (instead of Line 26) where propensity score matching was mentioned.

Study design: more details on urban and rural catchment areas should be provided.

Statistical analysis: Some details on what are propensity score and Rubin's B and how are they calculated should be provided.

A map showing the areas of study would be nice to add.

Author Response

Response to Reviewer 1 IJERPH 1728230

Reviewer 1 comments

Response to Reviewer 1 comments

Abstract: PSM should be used on Line 23 (instead of Line 26) where propensity score matching was mentioned.

We are thankful to the reviewer for picking up this mistake and we added the acronym line 23.

Study design: more details on urban and rural catchment areas should be provided.

We added one sentence to describe catchment areas p. 3 l. 117-118 and p. 4 l.119-120):

“Catchment areas are composed of villages or peri-urban neighborhoods called mahals where recruited CBR workers live and from where the program expands progressively to include nearby villages until covering extensively each district of the 13 included provinces.”

Statistical analysis: Some details on what are propensity score and Rubin's B and how are they calculated should be provided.

We added to the revised manuscript a definition of the propensity score matching as follows (p. 6 l. 256-259):

“Propensity score matching allows to reduce selection bias linked to the choice of a quasi experimental study design by controlling for selection on observables that might influence the probability of being in the CBR program intervention or control group [1].”

We also modified the revised manuscript to define Rubin’s B and R coefficients (p.6 l.268-272).

“In all analyses, Rubin’s B —which reflects the absolute standardized difference of the means of the propensity score in the treated and control groups— is below 25%, Rubin’s R —the ratio of the treated to control variances of the propensity scores— is within 0.5 and 2 and the percentage bias is below 10% for all covariates (Figures 2 and 3) [2].”

A map showing the areas of study would be nice to add.

The reviewer is right to suggest this addition and we added a map showing the areas where the study took place p. 3 l.108-109, Figure 1)

Reference mentioned

  1. Rosenbaum, P.R. and D.B. Rubin, The central role of the propensity score in observational studies for causal effects. Biometrika, 1983. 70(1): p. 41-55.
  2. Rubin, D.B., Using propensity scores to help design observational studies: application to the tobacco litigation. Health Services and Outcomes Research Methodology, 2001. 2(3-4): p. 169-188.

Reviewer 2 Report

The first survey date is cited as 2012 and covers the years up to 2016.
since the first surveys are almost 10 years ago, I would find a sentence on socio-political changes, especially after 2021 inevitable.

The research design is very extensive in terms of variables and the CBR - program shows to be very important.

To fight ingrained cultural stereotypes (like considering disabled as the consequence of a curse of God, spirits, jinn, fate (kismet) or black magic, 
would be interesting to know in depth in another article.

Author Response

Response to Reviewer 2 IJERPH 1728230

Reviewer 2 comments

Response to Reviewer 2 comments

The first survey date is cited as 2012 and covers the years up to 2016.
since the first surveys are almost 10 years ago, I would find a sentence on socio-political changes, especially after 2021 inevitable.

To address this important comment, we added a sentence at the end of the article (p. 22 l. 653-656):

Since the present study took place and to date, Swedish Committee for Afghanistan has continued to provide services to Afghan with disabilities and expand the home based disability program, covering more villages in the same 13 provinces [1].”

The research design is very extensive in terms of variables and the CBR - program shows to be very important.

To fight ingrained cultural stereotypes (like considering disabled as the consequence of a curse of God, spirits, jinn, fate (kismet) or black magic, would be interesting to know in depth in another article.

The reviewer is right and authors of the present study already published extensively about the stigma of persons with disabilities in Afghanistan. We referred to at least one of these studies in the present paper.

Reference mentioned

  1. Swedish Committee for Afghanistan. People with disabilities. 2022 [cited 2022 May 20, 2022]; Available from: https://swedishcommittee.org/our-work/people-disabilities/.

Reviewer 3 Report

Authors have proposed Access to services from persons with disabilities in Afghanistan: Is Community Based Rehabilitation making a difference?

the followings changes are required:

  1. Latest references should be added to this study.
  2. Improve the introduction section with relevant research.
  3. Fig. 1 (c), Fig. 2 (a), (e - g) should be redrawn with the same as other figures.
  4. The results section and Discussion parts can be merged with a section "Results and Discussions".
  5. Future work should be added in this manuscript.  

Author Response

Response to Reviewer 3 IJERPH 1728230

Reviewer 3 comments

Response to Reviewer 3 comments

  1. Latest references should be added to this study.

We look again into the existing literature on CBR programs, but also on disability and various issues (health, poverty, nutrition…) and we added new references.

We modify the introduction as follows (see p. 1, l.37-41):

Existing evidence points to higher levels of poverty [1, 2] and undernutrition [3], food insecurity and poor access [4], worse health status [5], scarce access to healthcare and rehabilitation services [6, 7], less availability of safe water and sanitation [8], lower access to quality education [9-12], poor employment opportunities [13-15] and higher risk of social and political exclusion [16-18].”

2.     Improve the introduction section with relevant research.

We modified the introduction to add recent research on CBR. The revised manuscript reads as follows (p. 1 l. 71-76):

Evaluation of CBR effectiveness remains scarce in LMICs making adoption of evidence-based practices problematic to achieve [19]. Many studies focus on a small sample of CBR participants and provide qualitative information on existing barriers and challenges to participation. Many quantitative studies evaluate the health component of CBR [20], a lot less education, few access to assistive devices [21, 22], nutrition [23], immunization [24], livelihoods [25] and social inclusion [26] and almost none empowerment [19].”

3.     Fig. 1 (c), Fig. 2 (a), (e - g) should be redrawn with the same as other figures.

We are not sure what the reviewer means by “with the same” but we redraw the figures mentioned using R software. All of them are done automatically by the software. We hope they match reviewer’s requirements. It will also be possible to tweak them at the time of publication.

4.     The results section and Discussion parts can be merged with a section "Results and Discussions".

We are not sure about the rationale for merging both results and discussion sections. Firstly, IJERPH instructions for authors indicate that there should be separated sections. We quote:

3. Results

This section may be divided by subheadings. It should provide a concise and precise description of the experimental results, their interpretation, as well as the experimental conclusions that can be drawn.”

“4. Discussion

Authors should discuss the results and how they can be interpreted from the perspective of previous studies and of the working hypotheses. The findings and their implications should be discussed in the broadest context possible. Future research directions may also be highlighted.”

5.     Future work should be added in this manuscript. 

We agree with the reviewer and we completed the conclusion by adding the following paragraph (see p. 22 l. 645-656):

“Future research is warranted to further define tools and mechanisms for standardization of CBR workers’ practice. Tools would include a new instrument used for monitoring the recruitment process of new participants and their progress. Standardization of practices would cover criteria for inclusion in ¾and implementation of¾ home-based therapy, advocacy, economic support (allocation of loans, vocational trainings). Another field of future research consists in assessing the impact of continuous capacity building of CBR staff. Finally, future research could examine greater participation of end beneficiaries and their families, as well as organization of persons with disabilities in the definition of services and activities provided by the CBR program. Since the present study took place and to date, Swedish Committee for Afghanistan has continued to provide services to Afghan with disabilities and expand the home based disability program, covering more villages in the same 13 provinces [27].”

Reference mentioned

  1. Mitra, S., The human development model of disability, health and wellbeing, in Disability, Health and Human Development. 2018, Springer. p. 9-32.
  2. Palmer, M., Disability and poverty: A conceptual review. Journal of Disability Policy Studies, 2011. 21(4): p. 210-218.
  3. Hume-Nixon, M. and H. Kuper, The association between malnutrition and childhood disability in low- and middle- income countries: systematic review and meta-analysis of observational studies. Tropical Medicine and International Health, 2018. 23(11): p. 1158-1175.
  4. Schwartz, N., R. Buliung, and K. Wilson, Disability and food access and insecurity: A scoping review of the literature. Health & place, 2019. 57: p. 107-121.
  5. Prynn, J.E. and H. Kuper, Perspectives on disability and non-communicable diseases in low-and middle-income countries, with a focus on stroke and dementia. International Journal of Environmental Research and Public Health, 2019. 16(18).
  6. Trani, J.-F., et al., Assessment of progress towards universal health coverage for people with disabilities in Afghanistan: a multilevel analysis of repeated cross-sectional surveys. The Lancet Global Health, 2017. 5(8): p. e828-e837.
  7. Mutwali, R. and E. Ross, Disparities in physical access and healthcare utilization among adults with and without disabilities in South Africa. Disability and Health Journal, 2019. 12(1): p. 35-42.
  8. Mactaggart, I., et al., Water, women and disability: Using mixed-methods to support inclusive wash programme design in Vanuatu. The Lancet Regional Health-Western Pacific, 2021. 8: p. 100109.
  9. Trani, J.F., et al., Assessment of progress in education for children and youth with disabilities in Afghanistan: a multilevel analysis of repeated cross-sectional surveys. Plos One, 2019.
  10. Bakhshi, P., G.M. Babulal, and J.F. Trani, Disability, Poverty and Schooling in Post-Civil War in Sierra Leone. European Journal of Development Research, forthcoming.
  11. Lamichhane, K., Disability and barriers to education: Evidence from Nepal. Scandinavian Journal of Disability Research, 2013. 15(4): p. 311-324.
  12. Mizunoya, S., S. Mitra, and I. Yamasaki, Disability and school attendance in 15 low-and middle-income countries. World Development, 2018. 104: p. 388-403.
  13. Tripney, J., et al., Interventions to improve the labour market situation of adults with physical and/or sensory disabilities in low‐and middle‐income countries: a systematic review. Campbell Systematic Reviews, 2015. 11(1): p. 1-127.
  14. Morwane, R.E., S. Dada, and J. Bornman, Barriers to and facilitators of employment of persons with disabilities in low-and middle-income countries: A scoping review. African Journal of Disability, 2021. 10.
  15. Trani, J.F., et al., Disability as deprivation of capabilities: Estimation using a large-scale survey in Morocco and Tunisia and an instrumental variable approach. Social Science and Medicine, 2018. 211: p. 48-60.
  16. Tobias, E.I. and S. Mukhopadhyay, Disability and social exclusion: Experiences of individuals with visual impairments in the Oshikoto and Oshana regions of Namibia. Psychology and Developing Societies, 2017. 29(1): p. 22-43.
  17. Tilly, L., Afraid to leave the house: issues leading to social exclusion and loneliness for people with a learning disability. Tizard Learning Disability Review, 2019.
  18. Trani, J.F., et al., Disability and Poverty in Morocco and Tunisia: A Multidimensional Approach. Journal of Human Development and Capabilities, 2015.
  19. Saran, A., H. White, and H. Kuper, Evidence and gap map of studies assessing the effectiveness of interventions for people with disabilities in low-and middle-income countries. Campbell Systematic Reviews, 2020. 16(1).
  20. Iemmi, V., et al., Community-based rehabilitation for people with physical and mental disabilities in low- and middle-income countries: a systematic review and meta-analysis. Journal of Development Effectiveness, 2016. 8(3): p. 368-387.
  21. Mauro, V., et al., The effectiveness of community-based rehabilitation programmes: an impact evaluation of a quasi-randomised trial J Epidemiol Community Health, 2014. 68(11): p. 1102-1108.
  22. Shore, S. and S. Juillerat, The impact of a low cost wheelchair on the quality of life of the disabled in the developing world. Medical Science Monitor, 2012. 18(9): p. CR533-CR542.
  23. Serin, G.E.Ç., S.S. Kisa, and Ö. Aydin, The effectiveness of nutrition and activity programmes for young adults with intellectual disabilities. International Journal of Caring Sciences, 2014. 7(2): p. 449.
  24. Soji, F., J. Kumar, and S. Varughese, Sustainability: lessons from a community-based rehabilitation programme in Karnataka, India. Knowledge Management for Development Journal, 2016. 12(2): p. 79-103.
  25. Zhang, G.F., et al., Integrated supported employment for people with schizophrenia in mainland China: a randomized controlled trial. The American Journal of Occupational Therapy, 2017. 71(6): p. 7106165020p1-7106165020p8.
  26. Trani, J.-F., J. Vasquez-Escallon, and P. Bakhshi, The impact of a community based rehabilitation program in Afghanistan: a longitudinal analysis using propensity score matching and difference in difference analysis. Conflict and Health, 2021. 15(1): p. 1-21.
  27. Swedish Committee for Afghanistan. People with disabilities. 2022 [cited 2022 May 20, 2022]; Available from: https://swedishcommittee.org/our-work/people-disabilities/.
